# Renewable formate from sunlight, biomass and carbon dioxide in a photoelectrochemical cell

Yuyang Pan[1,5], Huiyan Zhang [1,5] ✉, Bowen Zhang[1], Feng Gong[1], Jianyong Feng[2], Huiting Huang [2], Srinivas Vanka[3], Ronglei Fan[4], Qi Cao[1], Mingrong Shen [4], Zhaosheng Li [2] ✉, Zhigang Zou [2], Rui Xiao [1] & Sheng Chu [1] ✉

The sustainable production of chemicals and fuels from abundant solar energy and renewable carbon sources provides a promising route to reduce climate-changing $CO_2$ emissions and our dependence on fossil resources. Here, we demonstrate solar-powered formate production from readily available biomass wastes and $CO_2$ feedstocks via photoelectrochemistry. Non-precious NiOOH/α-$Fe_2O_3$ and Bi/GaN/Si wafer were used as photoanode and photocathode, respectively. Concurrent photoanodic biomass oxidation and photocathodic $CO_2$ reduction towards formate with high Faradaic efficiencies over 85% were achieved at both photoelectrodes. The integrated biomass-$CO_2$ photoelectrolysis system reduces the cell voltage by 32% due to the thermodynamically favorable biomass oxidation over conventional water oxidation. Moreover, we show solar-driven formate production with a record-high yield of 23.3 μmol $cm^{-2}$ $h^{-1}$ as well as high robustness using the hybrid photoelectrode system. The present work opens opportunities for sustainable chemical and fuel production using abundant and renewable resources on earth−sunlight, biomass and $CO_2$.

With the depletion of fossil resources and increasing concerns about greenhouse gas emissions, it is imperative to develop green approaches to access chemicals and fuels from renewable carbon resources instead of fossil feedstocks[1–3]. Solar-powered photoelectrochemical (PEC) $CO_2$ conversion provides an appealing strategy for producing sustainable chemicals/fuels and mitigating $CO_2$ emissions simultaneously[4,5]. In a typical PEC cell, cathodic $CO_2$ reduction reaction (CO2RR) is coupled to the anodic oxygen evolution reaction (OER), which requires a large overpotential and generates $O_2$ byproduct bearing low economic value[6,7]. Recent thermodynamic analysis has shown that more than 90% of the overall energy

requirements for $CO_2$ electrolysis to HCOOH stem from the OER[8]. Alternatively, OER can be substituted by other oxidation reactions that are thermodynamically more favorable and economically more feasible[9–16]. A common strategy is to replace OER with the oxidation of small organic substrates (function similar as sacrificial electron-donor agents), such as alcohols[17,18], aldehydes[19], urea[20], and 5-hydroxymethylfurfural[21]. For example, Verma et al. reported that the anodic glycerol oxidation to substitute OER could reduce the electricity consumption of electrolytic $CO_2$ reduction by up to 53%[22]. Although the concept of coupling organic oxidation reaction with CO2RR is promising, pure organics are usually costly and produced

[1]Key Laboratory of Energy Thermal Conversion and Control of Ministry of Education, School of Energy and Environment, Southeast University, Nanjing 210096, China. [2]Collaborative Innovation Center of Advanced Microstructures, National Laboratory of Solid State Microstructures, College of Engineering and Applied Sciences, Nanjing University, Nanjing 210093, China. [3]Department of Electrical and Computer Engineering, McGill University, 3480 University Street, Montreal, QC H3A 0E9, Canada. [4]School of Physical Science and Technology, Jiangsu Key Laboratory of Thin Films, Collaborative Innovation Center of Suzhou Nano Science and Technology, Soochow University, Suzhou 215006, China. [5]These authors contributed equally: Yuyang Pan, Huiyan Zhang. ✉e-mail: hyzhang@seu.edu.cn; zsli@nju.edu.cn; schu@seu.edu.cn

at a relatively small scale, which limit the economic feasibility and scalability to pair with CO2RR.

Biomass, an abundant, sustainable and low-cost natural carbon resource with an annual yield of 170 billion metric tons, is an ideal oxidation substrate to couple with cathodic CO2RR[23–25]. However, to our knowledge, the coupling of biomass oxidation reaction (BOR) with CO2RR in a PEC system has not yet been demonstrated. The development of such a PEC system has been beset with challenges including the convolution in simultaneous management of optical, electrical, and catalytic properties via a direct semiconductor-liquid junction, and the mismatch between anodic BOR with cathodic CO2RR to achieve efficient paired biomass-$CO_2$ photoelectrolysis. An additional challenge arises from the rigid and complex polymeric structure of biomass to obtain efficient and selective biomass conversion into single high-value product.

Herein, we report the integration of photoanodic biomass upgrading with photocathodic $CO_2$ reduction using photoelectrochemistry. Formate, a commodity chemical conventionally derived from fossil feedstocks via high temperature and pressure processes[26,27], was produced at high Faradaic efficiencies (FEs) of 91% and 85.2% at ambient conditions from photoanodic BOR and photocathodic CO2RR, respectively. A successive C1–C2 bond cleavage mechanism was revealed for the highly selective biomass conversion. The cell voltage of paired BOR-CO2RR photoelectrolysis was reduced by 32% relative to conventional OER-CO2RR photoelectrolysis. Moreover, solar-powered formate generation from biomass and $CO_2$ with an unprecedented formate production efficiency and high robustness was demonstrated.

## Results

### Design of the tandem PEC system

The designed PEC system is composed of two compartments of photoanodic biomass oxidation and photocathodic $CO_2$ reduction, as shown in Fig. 1. In the tandem configuration, photons with energies less than the bandgap of front absorber are transmitted and harvested by the back absorber, thus extending the solar spectrum utilization. In addition, the dual-absorber tandem devices can provide larger photovoltage than a single photoelectrode to drive the redox reaction with reduced electricity consumption. Significantly, compared with the traditional OER-CO2RR photoelectrolytic systems, the current BOR-CO2RR paired photoelectrolysis offers the following distinct advantages. First, the hybrid PEC system has excellent scalability and economic viability on both photoanode and photocathode using abundant biomass waste and $CO_2$ as the feedstocks. Second, the

voltage input is significantly reduced due to the less thermodynamically demanding oxidation of biomass compared to water, hence increasing the energy conversion efficiency. Third, the same high-value chemical commodity (i.e., formate) is produced at both anode and cathode simultaneously, minimizing cross-over reactions and reducing costly separation steps.

### Photoanode synthesis and characterization

Hematite ($\alpha$-$Fe_2O_3$) was chosen as the photoanode material due to its appropriate bandgap of ~2.1 eV as top absorber (up to a wavelength around 600 nm), low cost, nontoxicity, earth abundance and high photochemical stability[28,29]. The $\alpha$-$Fe_2O_3$ film was prepared by a facile spin-coating method onto fluorine-doped tin oxide (FTO) substrate followed with thermal annealing (see Methods for details). The as-prepared $\alpha$-$Fe_2O_3$ film shows a uniform orange color and a worm-like morphology in nanometric scale as observed by top-view scanning electron microscopy (SEM) (Fig. 2a). The side-view SEM image reveals the thickness of $\alpha$-$Fe_2O_3$ film is around 200 nm (Supplementary Fig. 1). X-ray diffraction (XRD) pattern, Raman spectra and X-ray photoelectron spectroscopy (XPS) analysis indicate the formation of pure hematite phase with predominant (110) facet (Supplementary Figs. 2–4). The bandgap of $\alpha$-$Fe_2O_3$ was estimated to be approximately 2.1 eV from UV-vis spectrum (Supplementary Fig. 5).

### Photoanodic glucose oxidation and cocatalyst screening

Glucose, a principal biomass-derived platform chemical, was selected as the modeling compound to screen and optimize the photoelectrode. Photoanodic glucose oxidation reaction (GOR) tests were conducted in Ar-purged 1 M aqueous solution of KOH (pH-13.6) under air mass 1.5 global (AM 1.5G) one-sun illumination (100 mW cm$^{-2}$) in a conventional three-electrode PEC cell. $\alpha$-$Fe_2O_3$-based sample, Ag/AgCl and Pt wire were employed as the working electrode, reference electrode and counter electrode, respectively. All the potentials are reported versus reversible hydrogen electrode (RHE) unless otherwise specified. OER is the major competing reaction with GOR in the aqueous electrolyte. Therefore, the photoanodic GOR and OER over $\alpha$-$Fe_2O_3$ are first compared by linear sweep voltammetry (LSV) curves, as shown in Fig. 2b. In the absence of glucose, bare $\alpha$-$Fe_2O_3$ photoanode displayed a catalytic onset potential of 1.0 V for OER. When glucose was introduced, the onset potential was obviously shifted negative by 300 mV to 0.7 V, indicative of thermodynamically more favorable GOR than OER. However, the photocurrent density was only slightly enhanced with the presence of glucose, indicating the sluggish reaction kinetics of glucose oxidation on unmodified $\alpha$-$Fe_2O_3$ surface.

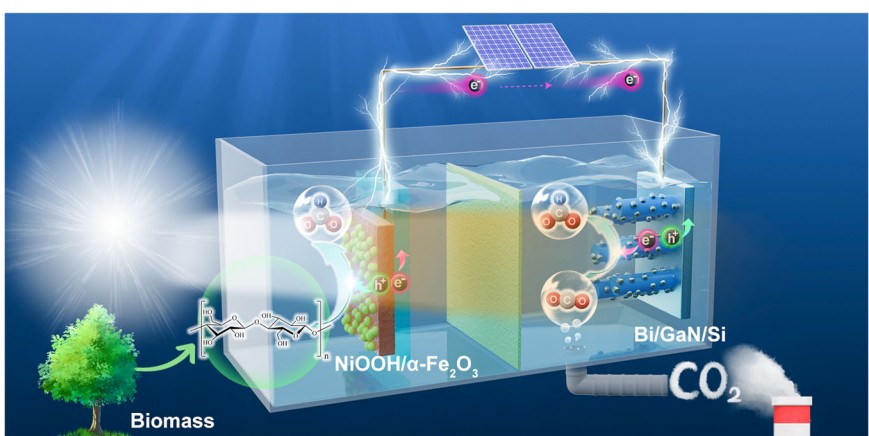

**Fig. 1 | Schematic illustration of the tandem PEC cell for formate production from biomass and CO2.** The PEC reactor consists of two compartments of photoanodic biomass oxidation (left-hand side) and photocathodic CO2 reduction (right-hand side) for concurrent formate production.

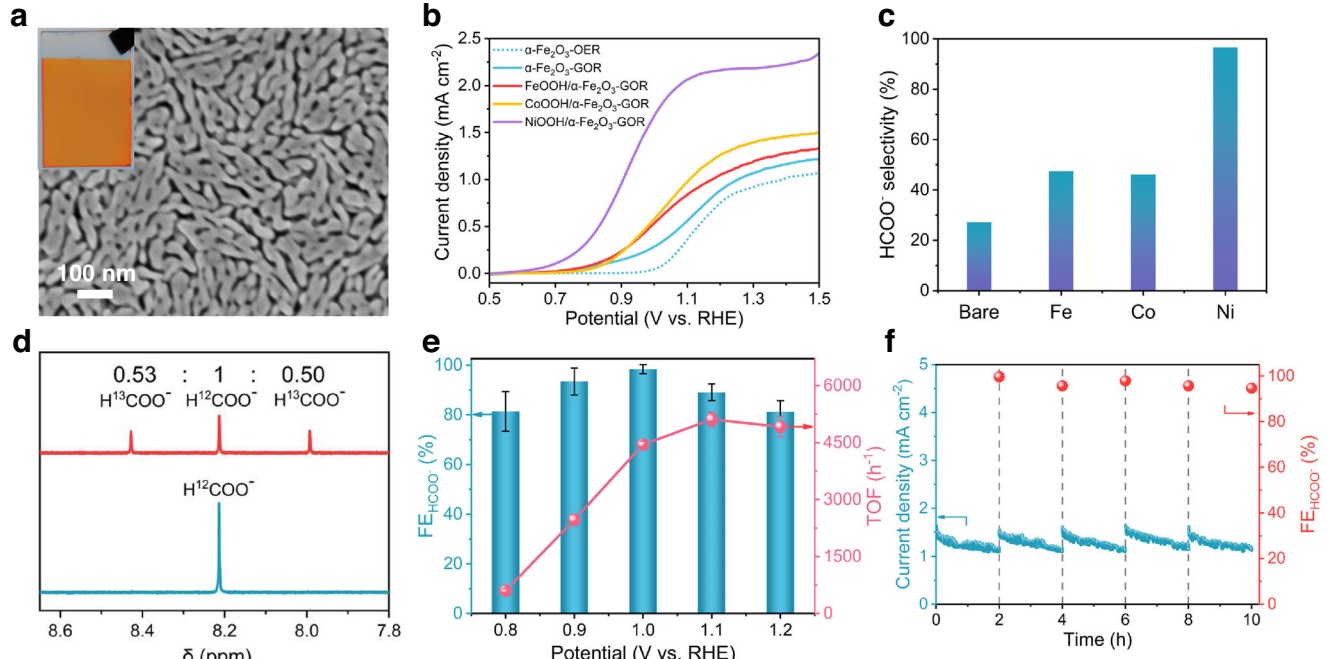

**Fig. 2 | Photoanodic glucose oxidation. a** Top-view SEM image and photograph (inset) of α-Fe2O3 photoanode. **b** LSV curves of α-Fe2O3 with different cocatalysts. **c** HCOO⁻ selectivity at 1 V for 2 h over α-Fe2O3 with different cocatalysts. **d** ¹H NMR spectra of the solution after 2 h PEC reaction over NiOOH/α-Fe2O3 at 1 V using 5 mM 13C-glucose + 5 mM 12C-glucose (red line) and 10 mM 12C-glucose (blue line) as reactants. **e** FEs and TOFs for HCOO⁻ production at varied potentials over NiOOH/

α-Fe2O3. **f** Chronoamperometry data and FEs for HCOO⁻ production at 1 V over NiOOH/α-Fe2O3. The dashed lines indicate the cleaning of photoelectrode and refreshing of electrolyte. All the experiments were conducted in Ar-purged 1 M aqueous solution of KOH (pH-13.6) with 10 mM glucose under AM 1.5G one-sun illumination. The error bars represent one standard deviation of three independent measurements.

To improve the GOR kinetics, a series of non-precious 3d transition metal oxyhydroxide MOOH (M = Fe, Co, and Ni), widely investigated as biomass oxidation electrocatalysts[30–32], were used to modify α-Fe₂O₃ photoanode. Fe-, Co-, and Ni-based hydroxides cocatalysts were respectively loaded on α-Fe₂O₃ by the same photodeposition procedure except different metal nitrate precursors (see Methods for details). Transmission electron microscope (TEM) analysis indicated that amorphous MOOH layers of 3–5 nm thickness were loaded on α-Fe₂O₃ (Supplementary Figs. 6–8). XPS analysis confirmed the $Fe^{III}$, $Co^{III}$ and $Ni^{III}$ states from FeOOH, CoOOH and NiOOH, respectively (Supplementary Figs. 9–11). Among the three cocatalysts, NiOOH modification exhibited far superior PEC performance in terms of photocurrent density and onset potential. A favorable onset potential of 0.6 V was achieved and the photocurrent density reached 1.7 mA cm⁻² at 1.0 V over NiOOH/α-Fe₂O₃, which was 6 times higher than that of unmodified α-Fe₂O₃ (0.28 mA cm⁻²). Such a drastic enhancement can be attributed to the enhanced electron-hole separation and accelerated glucose oxidation reactivity after NiOOH modification. Electrochemical impedance spectroscopy (EIS) Nyquist plot of NiOOH/α-Fe₂O₃ exhibits the smallest arc radius among the samples, indicating the highest electrical conductivity and interfacial charge transfer capability (Supplementary Fig. 12). It is noted that the photocurrent density increases rapidly with the increase of glucose concentration, indicating the responsive role of glucose substrate for the photocurrent (Supplementary Fig. 13).

To identify glucose oxidation products, high-performance liquid chromatography (HPLC) and NMR analysis of the electrolyte were performed. Chronoamperometric measurements were conducted by applying a constant potential of 1.0 V, at which no appreciable water oxidation could occur and thus the competitive OER was suppressed (Supplementary Fig. 14). Among the samples, NiOOH/α-Fe₂O₃ exhibited the highest formate selectivity of 96% for 2 h reaction (~98% FE considering 12-electron transfer of glucose oxidation to formate),

whereas bare α-Fe₂O₃, FeOOH/α-Fe₂O₃ and CoOOH/α-Fe₂O₃ samples displayed a poor selectivity less than 50% towards formate (Fig. 2c). Other products detected in the electrolyte were C2–C6 aldonic acids included glycolic acid ($C_2H_4O_3$), glyceric acid ($C_3H_6O_4$), erythric acid ($C_4H_8O_5$), arabic acid ($C_5H_{10}O_6$), gluconic acid ($C_6H_{12}O_7$), and glucaric acid ($C_6H_{10}O_8$), as shown in Supplementary Fig. 15. The total carbon balance approached almost unity for all the samples, and no CO₂ formation was detected in the reaction system (Supplementary Fig. 16). However, only NiOOH/α-Fe₂O₃ exhibited superior and unique catalytic performance for complete C–C bond cleavage of glucose into formate among the samples. The presence of formate products was confirmed by analyzing the electrolyte using ¹H NMR spectroscopy, showing a signal peak of 8.2 ppm corresponding to formate (Supplementary Fig. 17). No formate product was detected without light or external circuit, indicating the reaction proceeds via photoelectrocatalysis. A control experiment using formate as the reactant substrate indicated that formate was stable and cannot be overoxidized in the PEC reaction system (Supplementary Fig. 18). To investigate the origin of formate production, an isotope experiment was performed in which 5 mM ¹³C-glucose and 5 mM ¹²C-glucose were mixed for reaction. As shown in ¹H NMR spectra (Fig. 2d), the peak area ratio of H¹³COO⁻ to H¹²COO⁻ obtained is 1.03, which is very close to the theoretical value of 1. In addition, ¹³C NMR spectra also confirmed the presence of H¹³COO⁻ after PEC reaction (Supplementary Fig. 19). Furthermore, when the reaction was carried out in a PEC system in the absence of glucose, no formate was detected after the reaction. These results clearly indicate that the carbon source of formate originates from glucose.

The influence of the applied potentials on FE of formate over NiOOH/α-Fe₂O₃ was also investigated (Fig. 2e). The FE exhibited a volcano-like trend as a function of the applied bias and showed a maximum of 98% at 1 V. The lower FE in the low bias region was ascribed to the incomplete C–C bond cleavage because the remaining products were detected as C2–C6 aldonic acid. As the potential higher

than 1 V, OER competed and thus decreased the FE towards formate. To evaluate the accurate activity of NiOOH/α-Fe$_2$O$_3$, the turnover frequency (TOF) for formate was calculated and shown in Fig. 2e. A maximum TOF of ~5110 h$^{-1}$ was achieved at 1.1 V. It is worth noting that the loading amount of NiOOH had a significant effect on the PEC performance (Supplementary Fig. 20). The maximum activity and selectivity of glucose oxidation to formate was achieved with an optimum content of NiOOH that balances sufficient catalytic sites and effective light harvesting by underlaying α-Fe$_2$O$_3$. The durability of the NiOOH/α-Fe$_2$O$_3$ photoanode for glucose oxidation was also investigated (Fig. 2f). During the five runs of 10 h operation, the photoelectrode showed similar behaviors with stable photocurrent and high FEs of 97 ± 2% for formate production, indicating the high stability of NiOOH/α-Fe$_2$O$_3$ for GOR.

### The reaction mechanism and theoretical calculations

The reaction mechanism of glucose oxidation to formate was also investigated. HPLC analysis was used to track the dynamic evolution of glucose oxidation products over NiOOH/α-Fe$_2$O$_3$ as a function of reaction time (Supplementary Fig. 21). Aldonic acids from C1 to C6 were detected, indicative the successive C−C cleavage of glucose to formate. Besides the reaction pathways of aldonic acid intermediates, aldoses/aldehydes have been reported as possible intermediate species in the alcohol-to-formate transformation[33–35]. Therefore, we performed the comparative photoelectrochemical and kinetic evaluation using the observed aldonic acids and corresponding aldoses as reactants. As shown in Supplementary Fig. 21, the reaction rates of these substrates follow the order: aldoses > glucose > aldonic acids. Moreover, all aldoses can be oxidized to formate with nearly 100% FEs, which are significantly higher than the FEs of aldonic acids to formate (<50%). On the basis of the above results and the related knowledge in the literatures[36,37], we proposed the possible mechanism that glucose oxidation to formate proceeded mainly through an intermediate pathway of aldoses (Supplementary Fig. 22). Glucose was initially oxidized to arabinose and formate through C1−C2 bond cleavage. Then, arabinose underwent similar successive C1−C2 bond cleavages to form corresponding aldoses with one carbon atom less, and finally to formate. Meanwhile, aldoses oxidization to corresponding aldonic acids followed by C−C bond cleavages to formate was a minor reaction pathway.

To gain insight into the high activity and selectivity of NiOOH for glucose oxidation to formate at molecular level, density functional theory (DFT) calculations were conducted. The atomic models of NiOOH, CoOOH and FeOOH are shown in Supplementary Fig. 23. The Gibbs free energy diagrams of GOR on NiOOH, CoOOH and FeOOH are shown in Supplementary Fig. 24. The Gibbs free energy barrier for the conversion of glucose to formate over NiOOH is smaller than those over CoOOH and FeOOH, indicating a more energetically favorable reaction process. This theoretical result is consistent with the aforementioned experimental observations of extra high efficiency and selectivity for formate production on NiOOH.

### Raw biomass conversion

On the basis of the above findings, we aim to demonstrate the proof-of-concept transformation of real-world raw biomass, exemplified by using poplar sawdust, straw and bamboo. Due to the rigid polymeric structures and complex components of raw biomass (three main components: cellulose, hemicellulose and lignin), a facile acidic pretreatment strategy was adopted to depolymerize cellulose and hemicellulose to water-soluble sugar fragments, with insoluble lignin component filtered and removed (Fig. 3a). HPLC analysis of the resulting solutions show the generation of similar decomposition products of monosaccharide sugars including glucose and xylose with a (hemi)cellulose-to-glucose/xylose yield of 64 ± 1% (Supplementary Table 1, Supplementary Fig. 25).

The biomass-derived sugar solutions were subsequently used for PEC tests over NiOOH/α-Fe$_2$O$_3$ photoanode (Fig. 3b). Compared with the PEC performance in 1 M KOH electrolyte (water oxidation), substantially enhanced photocurrent responses and lowered onset potentials were achieved due to the introduction of biomass-derived solutions, suggesting the favorable BOR over WOR. Remarkably, the formate FEs adopting pretreated solutions from different biomass substrates were all higher than 90%, indicating the versatility of converting real-world raw biomass waste into formate with high selectivity (Fig. 3c). As a comparison, cellulose and hemicellulose were also applied with the same acidic pretreatment. The PEC performance of these substrates follows the order: glucose > (hemi)cellulose > raw biomass. The slightly lower PEC performance of biomass and (hemi) cellulose oxidation than glucose oxidation is probably ascribed to the presence of disaccharide and polysaccharide in the pretreated solution, which is due to the incomplete depolymerization of glycosidic bond in (hemi)cellulose structure. The LSV curves show that the PEC activities of glucose and xylose are comparable, whereas disaccharide has much lower performance (Supplementary Fig. 26). It is noted that the pretreatment step plays a critical role in the formate production, as control experiments show that the formate productivity without pretreatment is one or two orders of magnitude lower than those for pretreated samples (Supplementary Fig. 27).

Further, the operating stability of NiOOH/α-Fe$_2$O$_3$ was evaluated using pretreated solution from poplar sawdust at a constant potential of 1 V. As shown in Fig. 3d, a stable photocurrent is maintained with high formate FE of ~90% by ten successive cycles of 100 h, indicating the robust durability of NiOOH/α-Fe$_2$O$_3$ for BOR. This is further corroborated by post-PEC characterizations of sample including XRD, Raman and XPS analysis (Supplementary Fig. 28). The performance decreased with time in each cycle was likely due to the coverage of catalytic active sites by the adsorbed organic intermediate formed during the C−C bond cleavage process, which can be restored in the next run after the cleaning of photoelectrode surface. There was only a slight depletion of sugar concentration in each cycle (~9% concentration decrease from 10 mM glucose). Such a depletion did not have a large influence on the photocurrent decrease. Additionally, a small leaching of NiOOH cocatalyst to the electrolyte (~25% amount) was detected after the reaction in the preliminary stability test according to inductively coupled plasma−atomic emission spectroscopy (ICP-AES) analysis. Therefore, NiOOH cocatalyst was redeposited after 5$^{th}$ cycle to maintain the high cycle stability and activity. To avoid the leaching of NiOOH cocatalyst, developing novel cocatalyst deposition methods or stable morphological structure with an additional coating (i.e., core-shell structure) are necessary in follow-up works.

### Photocathodic CO$_2$ reduction

Bi nanoparticles modified GaN nanowire arrays on p-n junction Si wafer (denoted as Bi/GaN/Si) was adopted as the photocathode for CO2RR (Fig. 4a). GaN/Si photocathode takes advantage of the broad-band light absorption of Si (1.1 eV), and efficient electron extraction/ transportation effect as well as large surface area provided by GaN nanowires, which emerges as an excellent platform to achieve high performance CO2RR[38–40]. Bi/GaN/Si sample was fabricated using molecular beam epitaxy growth of GaN nanowires on Si, followed by facile electrodeposition of Bi nanoparticles (see Methods for details, Supplementary Fig. 29). The side-view SEM image (Fig. 4b) indicates that well-defined GaN nanowires with an average length of 350 nm and diameter of ~40 nm are vertically aligned to the Si substrate. High-resolution transmission electron microscopy (HRTEM) image shows Bi nanoparticles with an average size of 5 nm are well distributed across the GaN nanowires (Fig. 4c). The scanning transmission electron microscopy-energy dispersive X-ray spectroscopy (STEM-EDX) elemental mapping of single nanowire confirmed the uniform and conformal coverage of nanowire with Bi nanoparticles (Supplementary

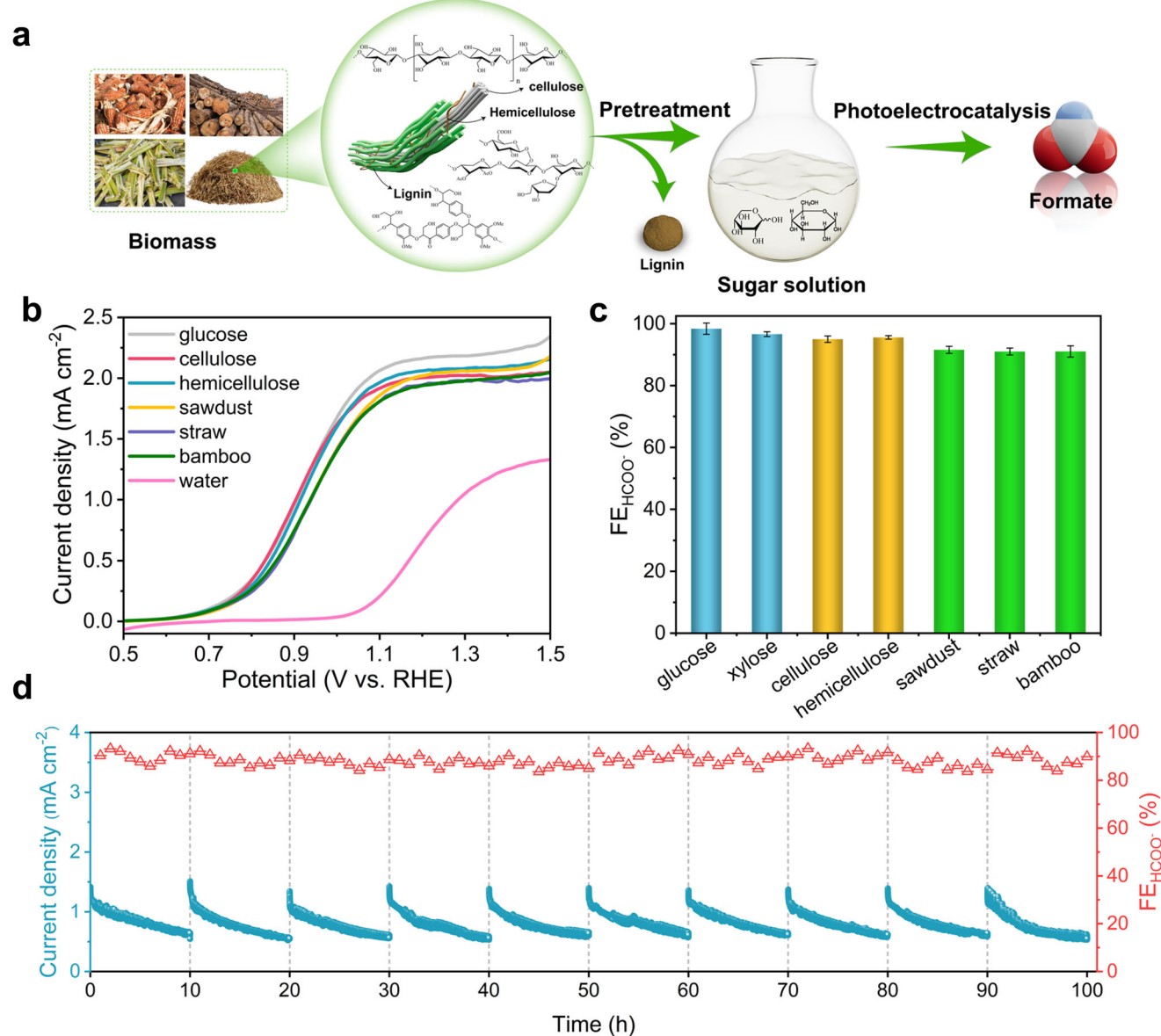

**Fig. 3 | Raw biomass conversion. a** Schematic illustration for the formate production from raw biomass. **b** LSV curves of different substrates over NiOOH/α-Fe2O3. **c** HCOO⁻ FEs of different substrates at 1 V for 2 h over NiOOH/α-Fe2O3. **d** Chronoamperometry data and FEs for HCOO⁻ production at 1 V in sawdust-derived sugar solution over NiOOH/α-Fe2O3 for 100 h. The dashed lines indicate cleaning of photoelectrode and refreshing of electrolyte. NiOOH cocatalyst was reloaded after the 5th cycle. All the experiments were conducted in Ar-purged 1 M aqueous solution of KOH (pH-13.6) under AM 1.5G one-sun illumination. The error bars represent one standard deviation of two independent measurements.

Fig. 30). The XPS spectra of Ga $2p_{3/2}$, N $1s$ and Bi $4f$ verified the chemical components of Bi/GaN/Si sample (Supplementary Fig. 31).

PEC performance of Bi/GaN/Si photoelectrode was investigated in $CO_2$-saturated 0.5 M $KHCO_3$ solution (pH = 7.5) in a conventional three-electrode cell under AM 1.5G one-sun illumination (100 mW cm⁻²). Compared to GaN/Si, Bi/GaN/Si shows a great enhancement with a favorable onset potential of 0.1 V and a high photocurrent density of 22.1 mA cm⁻² at −0.8 V (Fig. 4d), which are ascribed to the boosted reaction kinetics and enhanced charge carrier separation. Based on product analysis, $H_2$ was the primary product over bare GaN/Si (FE >98%). The incorporation of Bi cocatalyst drastically enhanced the selectivity towards formate production with a maximum FE of 85.2% at −0.2 V (Fig. 4e). Also, it was observed that the loading Bi amount had a great influence on the J-V behavior and product selectivity (Supplementary Fig. 32). It was found that the highest FE for formate was obtained at an optimum Bi amount that balanced the light absorbance

and catalytic activity sites. To our knowledge, Bi/GaN/Si reported in this work exhibited the lowest overpotential of 0 V to achieve high formate FE over 85% compared to other reported Si-based photocathodes (Supplementary Table 2). The excellent performance is attributed to the coupling effects of strong light harvesting of p-n Si (up to -1100 nm), efficient electron extraction and enhanced light trapping provided by GaN nanowires, and fast surface reaction kinetics of Bi cocatalysts. The stability evaluation of Bi/GaN/Si was carried out at a constant potential of −0.2 V for 10 h (Fig. 4f). Negligible photocurrent decrease can be detected and FE of formate maintains above 80%, indicating the high stability of Bi/GaN/Si. Furthermore, post catalysis SEM, TEM, and XPS analysis of Bi/GaN/Si photocathode confirmed the retention of nanowire morphology and chemical components (Supplementary Fig. 33). At last, a ¹³C isotopic experiment confirmed that formate product was originated from the reduction of $CO_2$ (Supplementary Fig. 34).

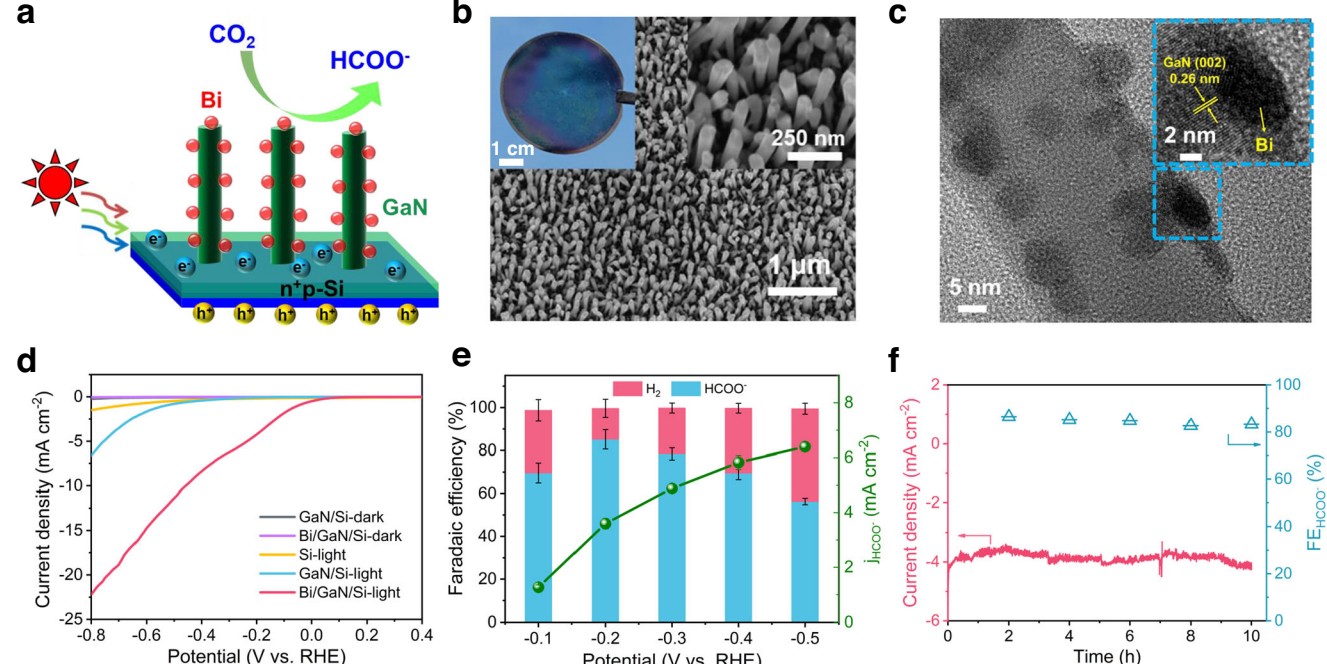

**Fig. 4 | Photocathodic CO2 reduction. a** Schematic illustration of Bi/GaN/Si photocathode for CO2 reduction to HCOO⁻. **b** Side-view SEM image and photograph (inset) of Bi/GaN/Si. **c** TEM image of Bi nanoparticles decorated GaN nanowire. **d** LSV curves. **e** FEs and jHCOO⁻ at varied applied potentials over Bi/GaN/Si.

**f** Chronoamperometric curve and corresponding HCOO⁻ FE at −0.2 V for 10 h over Bi/GaN/Si. All the experiments were conducted in CO2-saturated 0.5 M KHCO3 aqueous solution (pH-7.5) under AM 1.5G one-sun illumination. The error bars represent one standard deviation of two independent measurements.

### Biomass-CO₂ paired photoelectrolysis system

An integrated PEC cell with NiOOH/α-Fe₂O₃ as photoanode and Bi/GaN/Si as photocathode was assembled in a two-electrode tandem configuration (Fig. 5a). The longer-wavelength photons (λ > 600 nm) that are transmitted through the front α-Fe₂O₃ photoanode can be utilized by the rear Si-based photocathode, which allows the hybrid device to encompass a large portion of solar spectrum. J-V curves of the tandem PEC cell with and without biomass-derived pretreated solution are shown in Fig. 5b. Compared to the conventional OER-CO2RR system, the PEC performance of BOR-CO2RR is significantly enhanced in terms of onset potential and photocurrent density due to the thermodynamically favorable biomass oxidation over water oxidation. The cell voltages are reduced over 300 mV to attain the same photocurrent densities of 0.1, 0.2, 0.5, and 1 mA cm⁻² in the presence of pretreated biomass (Fig. 5c). And the cell voltage of biomass-CO₂ photoelectrolysis is reduced by 32 ± 2% with the introduction of pretreated biomass, in comparison to the system operating with water oxidation.

To achieve a self-powered PEC formate production, we integrated the PEC tandem cell with two pieces of commercial crystalline Si solar cells as the bottom absorber that could supply a photovoltage of ~1.2 V (positioned side-by-side with photocathode). In this tandem configuration, photons with energies less than the bandgap of the α-Fe₂O₃ are absorbed by the bottom Bi/GaN/Si photocathode and Si solar cell. The theoretical operating current of the integrated system was estimated to be 1.25 mA cm⁻², which was determined by overlaying the J−V curves of photoelectrodes and Si solar cell in the integrated system (Fig. 5d). Long-term test of four consecutive runs with each run of 20 h was conducted to evaluate the performance and durability of the device (Fig. 5e). After each cycle, the electrode was thoroughly cleaned by deionized water and the electrolyte was replaced with fresh solution. The integrated device can stably operate for 80 h with less than 15% loss in formate production yield, indicating the excellent reusability of the system. The PEC system produces formate simultaneously at both photoanode and photocathode in the absence of external bias with respective amounts of 1044 μmol cm⁻² and 817 μmol cm⁻² after

80 h (Supplementary Fig. 35). The average formate FEs were calculated to be as high as ~160%, with ~90% and ~70% for photoanode and photocathode, respectively. Overall, the PEC system produced formate at a high rate of 23.3 μmol cm⁻² h⁻¹ from biomass and CO₂ during 80 h operation. Significantly, the formate production performance of the device far excels the state-of-the-art solar formate production systems from the conventional OER-CO2RR photoelectrolysis (Fig. 5f, see detailed comparison in Supplementary Table 3). The high performance of our device can be ascribed to the replacement of OER with less energetically demanding BOR and the efficient concurrent formate production at both photoelectrodes. The solar to chemical energy conversion efficiency was calculated to be ~0.19% (Supplementary Note 1). Further improvement of the efficiency is anticipated by integrating other candidate photoelectrode with higher photocurrent at favorable potentials, such as BiVO₄ and SnS[41–44].

### Discussion

In summary, we have reported a co-photoelectrolysis strategy for concurrent formate production from biomass and CO₂ in a tandem PEC cell. High formate FEs of 91% and 85.2% were achieved from photoanodic biomass oxidation and photocathodic CO₂ reduction, respectively. Compared to the conventional OER-CO2RR photoelectrolysis, the cell voltage of paired BOR-CO2RR photoelectrolysis is reduced by 32%. Furthermore, solar formate production from biomass and CO₂ with an unprecedented production efficiency and high robustness has been demonstrated. This work provides a promising sustainable approach for the synthesis of chemical and fuels from abundant and renewable carbon feedstocks using sunlight as the only energy input.

### Methods
#### Photoanode synthesis
α-Fe₂O₃ photoanode was prepared via a facile spin-coating method. The α-Fe₂O₃ photoanode was doped with 3 at% Ti (optimized) to enhance the electronic conductivity and the charge separation

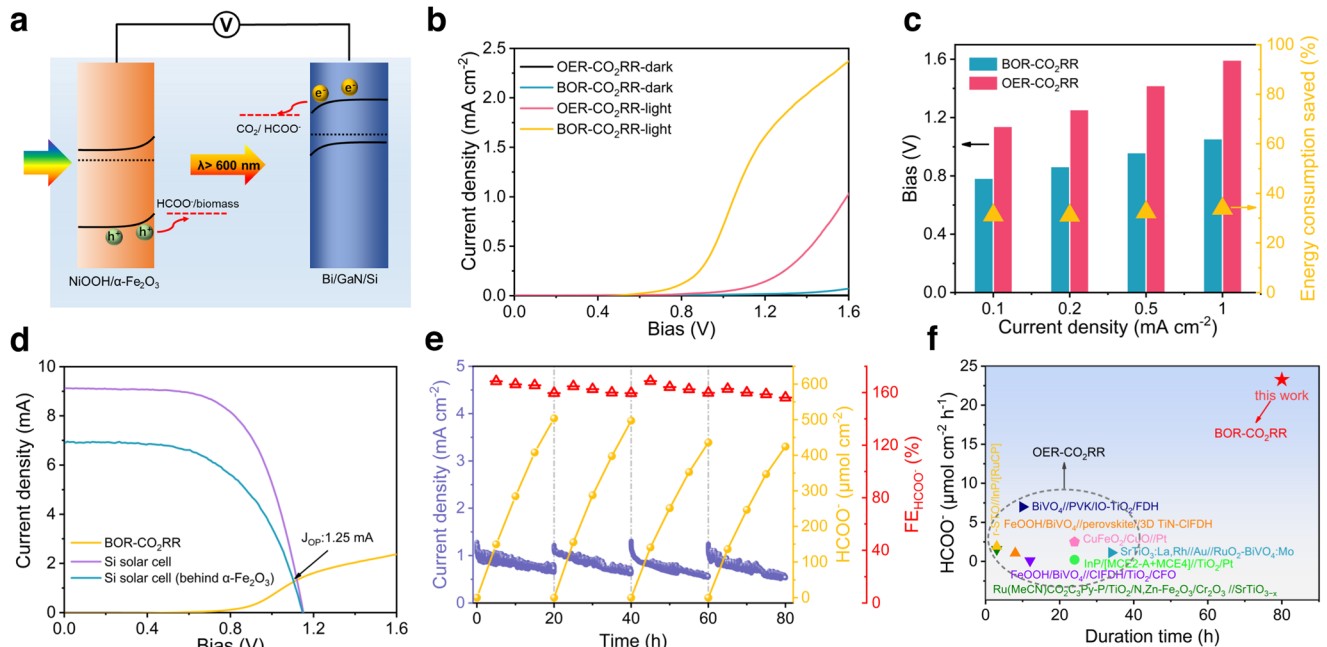

**Fig. 5 | Biomass-CO₂ paired photoelectrolysis system. a** Schematic illustration of two-electrode NiOOH/α-Fe₂O₃//Bi/GaN/Si tandem PEC cell for formate production from biomass and CO₂. **b** LSV curves of the tandem PEC cell with and without biomass addition. **c** Comparisons of the bias required to achieve varied current densities with and without biomass addition. **d** LSV curves of Si solar cell and its intersection with LSV curve of the tandem PEC cell. **e** Chronoamperometric curve and corresponding HCOO⁻ productivity and FEs of the integrated device. The dashed lines indicate cleaning of photoelectrodes and refreshing of electrolytes. **f** Performance comparison of this work (BOR-CO2RR) with state-of-the-art formate production from conventional OER-CO2RR photoelectrolysis. All the experiments were conducted under AM 1.5G one-sun illumination using poplar sawdust as the reactant substrate.

efficiency[45,46]. Firstly, FTO glass substrate was sequentially cleaned by ultrasonication in acetone, ethanol, and deionized water for 30 min each, followed by UV ozone cleaning for 10 min to remove surficial contaminants. Subsequently, a compact α-Fe₂O₃ layer was deposited on the cleaned FTO glass via spin-coating of a Fe₂O₃ precursor solution at 3000 rpm for 30 s and dried at 60 °C for 20 min, followed by annealed at 600 °C for 2 h in air. The precursor solution was formed by dissolving 1 mmol Fe(NO₃)₃ 9H₂O, 0.03 mmol bis(2,4-pentanedionato) bis(2-propanolato)titanium(IV) (TCI, 75% in isopropyl alcohol) and 0.2 mmol acetylacetone (Sigma-Aldrich, ≥99%) in 1 mL ethanol, and standing for 2 days.

NiOOH cocatalyst was loaded on α-Fe₂O₃ by a simple photo-deposition method. A piece of α-Fe₂O₃ photoanode (1 cm × 1.5 cm) was immersed in 1 mM Ni(NO₃)₂ aqueous solution (10 mL), and then 10 mM NaIO₃ was added to sever as electron sacrificial agent. The photo-reactor was subsequently evacuated for 10 min, and irradiated for 30 min using a 300 W Xe lamp for the photodeposition. CoOOH and FeOOH was photodeposited on α-Fe₂O₃ using the same procedure except for the use of Co(NO₃)₂ and Fe(NO₃)₂ as precursors.

### Photocathode synthesis

GaN nanowire arrays were grown on n⁺-p Si wafer by plasma-assisted molecular beam epitaxy under nitrogen rich condition as previously reported[47]. The growth temperature was 790 °C with ~1.5 h duration time. The Ga flux pressure was 6 × 10⁻⁸ Torr with a plasma power of 350 W. Bi cocatalyst was electrodeposited on GaN/Si by a cyclic vol-tammetry method. 0.25 mmol Bi(NO₃)₃ 5H₂O was dissolved in 50 mL ethylene glycol to sever as Bi precursor solution. The electrodeposition was carried out in a PEC chamber by a typical three-electrode configuration, employing saturated Ag/AgCl as reference electrode and Pt foil as counter electrode. The depositing step was realized by sweeping potential between −2.5 to +2.5 V with a scanning rate of 100 mV/s. The loading amount of Bi can be readily controlled by tuning the electrodeposition cycles.

### Material characterizations

XRD patterns of α-Fe₂O₃ films were performed on Bruker D8 Advance X-ray diffractometer (Bruker Company, Germany) equipped with a Cu Ka radiation source (λ = 1.54060 Å) at a scan rate of 2° min⁻¹. Raman spectra of α-Fe₂O₃ films were characterized with a confocal laser Raman spectrometer (Renishaw inVia) with a 532 nm laser excitation source. The SEM image was acquired on a Hitachi S-4800 system. TEM image was measured using a FEI Tecnai G2 F20 microscope. XPS analysis was performed on a Thermo Scientific Nexsa X-ray photoelectron spectrometer using monochromatized Al Kα radiation (1486.6 eV). ICP-AES analyses were conducted on an Agilent 725-ES instrument.

### Raw biomass pretreatment

Poplar sawdust, straw and bamboo were all obtained from Shandong Province, China. Raw biomass was firstly crushed and sieved to obtain powders with particle size of 60–80 meshes (0.18–0.25 mm). Then, they were dried in a blast drying oven at 105 °C for 8 h and stored for pretreatment. An acidic pretreatment method was adopted for the conversion of raw biomass substrates to soluble sugars as described previously[48]. Typically, 0.5 g of raw biomass substrate was added to 7.5 mL of 7.2 wt % H₂SO₄ solution. The mixture was vigorously stirred at room temperature for 2 h. Then, 90 mL of water was added and heated under reflux (100 °C) under magnetically stirring for 5 h in a round-bottom-flask. Afterwards, insoluble lignin was filtered and removed. The resultant solution was treated with Ba(OH)₂ to removed H₂SO₄. The formed BaSO₄ solid was removed by centrifugation. Before PEC test, KOH was added to adjust the pH of sugar solution to be ~13.6.

### Photoelectrochemical measurement

Photoelectrochemical measurements were conducted on an electro-chemical workstation (CHI 760E, CH Instruments) in a three-electrode system (Ag/AgCl electrode as reference electrode and Pt foil as counter electrode). Nafion proton exchange membrane was used to separate working electrode chamber from the counter electrode

chamber. A simulated solar irradiation (Newport Oriel, AM 1.5 G, 100 mW cm⁻²) was used for light illumination. The light intensity was calibrated to be 100 mW cm⁻² by the standard reference of a Newport 91150 V silicon cell before use. Unless otherwise stated, all the potentials reported were converted to RHE scale using the following equation: E(versus RHE) = E(versus Ag/AgCl) + 0.1976 + (0.0591× pH).

Photoanode oxidation and photocathode $CO_2$ reduction experiments were conducted in Ar-purged 1 M KOH aqueous electrolyte (pH-13.6) and $CO_2$-purged 0.5 M $KHCO_3$ aqueous electrolyte (pH-7.5), respectively. Before PEC measurement, Ar or $CO_2$ was purged through the electrolyte for 30 min and then the cell was sealed. Oxidation of glucose or other substrates experiments were conducted in the same manner except that the introduction of glucose or other substrates into the electrolyte. LSV data was recorded at a scan rate of 10 mV s⁻¹. EIS measurements were measured at a frequency range of 100,000 to 0.1 Hz with the amplitude of 10 mV. All the curves were measured without IR compensation. The temperature was maintained at ~278 K.

In the tandem PEC cell tests, a sealed H-type PEC cell was adopted with a bipolar membrane (Fumasep FBM) to separate anode chamber and cathode chamber. A home-made two-junction Si solar cells was connected with the tandem cell using a copper wire. The PEC cell was illuminated from the photoanode side. The light transmitted through the front α-$Fe_2O_3$ photoanode (1 cm²) was utilized by the rear Bi/GaN/Si photocathode (0.5 cm²) and Si solar cell (0.5 cm²).

## Product analysis and quantification

Liquid products were quantified by high performance liquid chromatography (HPLC, Shimadzu LC-2010) equipped with a refractive index detector and a UV detector ($\lambda = 210$ nm). A Shodex SUGAR SH1011 column was used for product separation. 5 mM $H_2SO_4$ was used as the mobile phase with a constant flow rate of 0.7 mL min⁻¹ at 50 °C. The identification of products and calculation of their concentrations were determined from calibration curves by applying standard solutions with known concentrations. Typically, 100 μL of electrolyte solution was taken out from PEC cell and diluted with 0.125 M of $H_2SO_4$ solution to adjust the pH to ~3, then 20 μL of the diluted solution was injected into HPLC.

¹H and ¹³C NMR spectra were recorded on Bruker Avance II 300 instruments. In each test, 800 μL of the sample solution was mixed with 200 μL $D_2O$, and 10 μL DMSO was employed as internal standard. It should be noted that alkaline anode solution was first neutralized with $H_2SO_4$ before adding $D_2O$ and DMSO. Presaturation method was used to suppress the water signal.

Gas-phase products were analyzed by a gas chromatography system (Shimadzu GC-2010) equipped with thermal conductivity detector and flame ionization detector connected to molecular sieve column and Porapak N column.

Selectivity of products was calculated as the molar ratio of specific product to consumed reactant:

$$\text{Selectivity}(\%) = \frac{\text{mol of specific product}}{\text{mol of consumed reactant}} \times 100\% \quad (1)$$

Faradaic efficiency (FE) was calculated using the following equation:

$$\text{FE}(\%) = \frac{N_{\text{product}}}{Q/ZF} \times 100\% \quad (2)$$

where $N_{\text{product}}$ is the moles of formed product, $Q$ is the total charge passed, $Z$ is the number of electron transfer to generate the product from reactants ($Z = 2$ for formate production from sugar or $CO_2$), $F$ is the Faraday constant (96485 C mol⁻¹).

Turnover frequency (TOF) for formate was calculated by:

$$\text{TOF}(h^{-1}) = \frac{N_{\text{HCOO}^-}}{\text{area of electrode} \times \text{loading density of cocatalyst} \times \text{time}} \quad (3)$$

## Computational details

DFT calculations were conducted using the Perdew-Burke-Ernzerhof (PBE) functional[49]. The projector augmented wave (PAW) method was employed to describe the electron-ion interaction[50]. The kinetic energy cutoff for plane wave was set to 450 eV. The Brillouin zone was sampled by Gamma grid with $2 \times 2 \times 1$ k-points. The criterion of structure relaxation was set to $10^{-5}$ eV for total energy and 0.05 eV Å⁻¹ for the force of each atom. The correction of vdWs force was performed by employing DFT-D3 method. Climbing nudged elastic band (CI-NEB) method was introduced to search for the transition states. The computational models were constructed by cleaving the (001) facet of NiOOH, CoOOH and FeOOH, containing 16 metal (Ni/Co/Fe) atoms, 16 H atoms and 32 O atoms. A 20 Å vacuum layer was introduced to avoid the interaction effect between different layers.

## Data availability

All the data that support the findings of this study are available from the corresponding authors on reasonable request. Source data are provided with this paper.

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

## Acknowledgements

This work was supported by the Scientific and Technological Innovation Project of Carbon Emission Peak and Carbon Neutrality of Jiangsu Province (BE2022024, BE2022028-4), the National Natural Science Foundation of China (51822604, 22005048, 22025202), and the Natural Science Foundation of Jiangsu Province (BK20200399). We thank Mr. Tao Chen, Prof. Yu Zhou, Prof. Jun Wang, and Dr. Ling Gao for help on the product analysis. S.C. would like to thank the support from the "Zhishan Young Scholar" Program of Southeast University. S.C. also acknowledges the fruitful discussions with Prof. Zetian Mi.

## Author contributions

S.C., H.Z. and Y.P. designed the project and analyzed the data. H.Z., Z.L. and S.C. co-supervised the work. Y.P. performed most of the experiments. B.Z. participated in the preparation, characterization and PEC test of sample. F.G. performed DFT calculations. J.F. and H.H.

contributed to photoanode preparation and characterization. S.V. contributed to photocathode synthesis. R.F. contributed to paired system construction. Q.C. contributed to electron microscopy analysis. M.S., Z.Z. and R.X. contributed to result analysis and discussions. The manuscript was written by S.C. and Y.P. with contributions from other co-authors.

## Competing interests

The authors declare no competing interests.
