## [Peer Review File · Nature Communications]

REVIEWER COMMENTS

Reviewer #1 (Remarks to the Author):

The m/s of Sheng Chu et al reports on the paired photoelectrolysis of CO₂ and biomass derived sugars. The key claim of the manuscript is the achieved high selectivity for formate production on both electrodes. I've read the revised manuscript very carefully, and I found that it contains new insights related to pairing two high-value photoelectrochemical processes, with potential interest of the solar energy conversion community. At the same time there are several overstatements in the manuscript, which should be revised. I also have some questions, as listed below. Overall, I think that a major revision is needed before this study could be published in Nature Communications.

Major comments

1. The concept of paired photoelectrolysis is not new. there are excellent review articles on this topic. Therefore the claim that "the present work opens new scenarios" is not really true. Coupling CO₂R with biomass oxidation is indeed an emerging field, but it is not true that there is no precedence. See for example: Nat. Synth. 2022, 1 (1), 77–86. Nat. Commun. 2020, 11 (1).
2. What is the exact selectivity definition the Authors use? What is the relationship between the 96% "formate selectivity" and the 98% FE?
3. Did the authors try to detect eventually forming CO₂ at the anode? It is not easy to detect quantitatively, but in such reactions there is always some CO₂ formation. In fact, it is very hard to believe that the Authors can account for the complete carbon balance without detecting the CO₂...
4. the Author claim that "the overall energy consumption of biomass-CO₂ photoelectrolysis is reduced by 32±2%, in comparison to the system operating with water oxidation" which does not take into account the energy need of biomass pretreatment, and the issues related to the generated chemical waste. What is the fate of the byproducts during the pre-conversion step? Apparently, the pretreatment is very important "control experiments show that the formate productivity without pretreatment is one or two orders of magnitude lower than those for pretreated samples" therefore it should be included in the complete picture.
5. What is the reason for the performance decrease with time, and why does the performance restore after replacing the electrolyte? Is this a depletion in the sugar?
6. There is a built in chemical bias because of the pH difference on the two sides. This also has to be taken into account during the energy efficiency calculations.
7. The "unassisted" operation is also somewhat misleading. The term unassisted is generally used when a PEC cell alone can drive both half reactions. This is not the case here, as two additional Si solar cells are coupled to the system, if I understand correctly.

8. I know that it is not straightforward to do, but a solar to chemical conversion shall be estimated for the whole system. Ultimately this could show the merit of the presented approach.

9. The comparison in Fig. 5f is totally unfair. The other studies involve different anode processes, than the present work. Thereby apples are being compared to oranges, Furthermore, the Figure legend "Performance comparison of this work with state-of-the-art unassisted PEC formate production from CO₂ and H₂O." is factually wrong, because "biomass" is also a reactant here.

Reviewer #2 (Remarks to the Author):

A novel scheme of biomass oxidation at anode coupled with CO₂ reduction at cathode (BOR-CO₂RR) using photoelectrochemistry to produce formate with Faradic efficiencies (FE) is reported. Non-precious 3d transition metal oxyhydroxide MOOH (M = Fe, Co, and Ni) supported on α -Fe₂O₃ photoanode and Bi/GaNi/Si-wafer photocathode was used. Authors have demonstrated a 32% reduction in energy consumption for BOR-CO₂RR compared to CO₂RR coupled with water oxidation. Plausible mechanism based on product analysis are postulated. DFT calculations are performed to support variation in photoanode activity among Fe, Co and Ni system.

This is an original study with proof-of-concept demonstration and conceptual design with potential for further development. The results are noteworthy, and authors have cited state-of-the-art developments in the field. Sufficient details of experimental studies are provided to reproduce the results. However, several issues remain to be addressed, notably morphological stability of photoanode (i.e., how to avoid leaching of NiOOH).

Brief DFT calculations presented in the manuscript doesn't necessarily establish high FE of Ni-anode system compared to Fe (FE of Ni-system is almost twice of Fe-system) as energy values associated with intermediated and TSs for these two systems are too close.

Authors have demonstrated order of magnitude improvement in performance of BOR-CO₂RR when biomass are pretreated. Pretreatment affects different biomass differently depending on composition. A suggested range of composition of substrate after pretreatment will be helpful to ensure target productivity from BOR-CO₂RR.

Restructuring some of the languages as indicated in the annotated manuscript is suggested.

Reviewer suggest accepting this manuscript for publication with minor edits/re-structuring as indicated above.

Reviewed by:

Amitava Sarkar, Ph.D.

Resident Visiting Scientist (Stanford University)

Corporate Research Scientist for North America (TotalEnergies SE)

Room 311, Shriram Center for Bioengineering & Chemical Engineering

Stanford University

443 Via Ortega, Stanford, CA 94305, USA

M: +(1) 650 304 9543

amitava@stanford.edu

amitava.sarkar@totalenergies.com

Reviewer #3 (Remarks to the Author):

The authors demonstrated a new photoelectrochemical system utilizing biomass oxidation reaction and CO₂ reduction reaction. Unifying the reaction product from both electrodes into formate was impressive and could appeal to the readers of the journal. However, some issues need to be addressed. Therefore, it is recommended to be accepted after major revision.

1. The current title and abstract don't include information on the materials used or, in this case, systematic details. As Nature Communication aims to present important advances of significance to specialists within each field, the title and abstract should be more specific.

2. Though the Solar-to-fuel or solar-to-chemical efficiency is not calculated in this article, an estimation over figure 5 shows that comparatively low STF was achieved in this article. Recently an STF value exceeding 11% was reported. Refer to small, 2021, 17, 29, 2101128. Please add calculated

STF values to the manuscript. Was the PV-PEC system optimized to achieve the best performance of formate production?

3. The faradaic efficiency (maximum 85.2%) of the photocathode is relatively low when compared with other reported performances of Bi/GaN NW electrodes even after cycle optimization, as shown in Figure S31. Maybe the 20-cycled sample should be tested at higher potentials.

4. The X-axis in figure S31(b) is wrongly labeled.

5. Is there any reason why hematite and GaN/Si were used as photoelectrode materials? There are several more candidates with higher photocurrent, such as BVO or SnS. Refer to the following [1] Energy Environ. Sci., 2022,15, 672-679 [2] Adv. Sci., 2021, 8, 21, 2102458 [3] ACS Appl. Mater. Interfaces 2020, 12, 13 [4] Solar RRL, 2019, 3, 12, 1900301

Point-by-point response to the reviewers' comments

Title: “Renewable formate from sunlight, biomass and CO₂”

(Manuscript ID: NCOMMS-22-36755-T)

We sincerely thank all Reviewers for their valuable comments and suggestions, which are certainly helpful in improving the quality of our work. We have carefully and systematically responded to all the points raised. The Reviewers' comments are in blue fonts and our responses in black fonts. We have also highlighted the revised text in yellow in the main text. Provided below are our detailed responses to each point.

Reviewer #1 (Comments to authors)

The m/s of Sheng Chu et al reports on the paired photoelectrolysis of CO₂ and biomass derived sugars. The key claim of the manuscript is the achieved high selectivity for formate production on both electrodes. I've read the revised manuscript very carefully, and I found that it contains new insights related to pairing two high-value photoelectrochemical processes, with potential interest of the solar energy conversion community. At the same time there are several overstatements in the manuscript, which should be revised. I also have some questions, as listed below. Overall, I think that a major revision is needed before this study could be published in Nature Communications.

Response:

We thank the reviewer very much for careful evaluation of our work, and stating that our work contains new insights and potential interest of the solar energy conversion community. We also greatly appreciate the reviewer for the constructive comments and suggestions to improve the quality of the work. We have revised the manuscript accordingly, and our point-by-point responses can be found below.

Comment 1:

The concept of paired photoelectrolysis is not new. there are excellent review articles on this topic. Therefore the claim that "the present work opens new scenarios" is not really true. Coupling CO₂R with biomass oxidation is indeed an emerging field, but it is not true that there is no precedence. See for example: Nat. Synth. 2022, 1 (1), 77 – 86. Nat. Commun. 2020, 11 (1).

Response:

We thank the reviewer for the comment. We agree with the reviewer that the concept of paired photoelectrolysis is not new. The major finding of our work is pairing CO₂R with biomass oxidation for simultaneously formate production with high selectivity, which is an emerging field. Therefore, we have revised the claim that "the present work opens new scenarios" to "the present work opens opportunities" in the Abstract of the revision (*Page 2, Line 11*). The relevant previous works including the two mentioned by the reviewer have been cited in the revision.

Comment 2:

What is the exact selectivity definition the Authors use? What is the relationship between the 96% "formate selectivity" and the 98% FE?

Response:

We thank the reviewer for the comments. Selectivity of products was calculated as the molar ratio of specific product to consumed reactant:

$$\text{Selectivity (\%)} = \frac{\text{mol of specific product}}{\text{mol of consumed reactant}} \times 100\%$$

Faradaic efficiency (FE) was calculated based on the equation below:

$$\text{FE (\%)} = \frac{\text{mol of electrons consumed for specific product}}{\text{mol of electrons passed}} \times 100\%$$

There is a positive correlation between the formate selectivity and FE. We have modified the selectivity definition in the Methods of the revised manuscript (*Page 19, Line 6*).

Comment 3:

Did the authors try to detect eventually forming CO₂ at the anode? It is not easy to detect quantitatively, but in such reactions there is always some CO₂ formation. In fact, it is very hard to believe that the Authors can account for the complete carbon balance without detecting the CO₂...

Response:

We thank the reviewer for the comment. We have tried to detect the possible CO₂ product by gas chromatography, as shown in Figure R1 below. The signal of CO₂ peak was not observed, indicating no formation of CO₂. This is consistent with the near-unity carbon balance detected in the system. In addition, a control experiment using formate as the reactant substrate showed that formate was stable in the reaction system (Figure R2). This indicates that formate cannot be over-oxidized to CO₂ in our system. We have added related discussions in the revised manuscript (*Page 7, Line 15*).

Figure R1. Gas chromatography spectrum of products after 2 h GOR at 1 V under AM 1.5G one-sun illumination over NiOOH/ α -Fe₂O₃.

Figure R2. | **a** LSV curves over NiOOH/ α -Fe₂O₃ in Ar-purged 1 M aqueous solution of KOH with and without formate added under AM 1.5G one-sun illumination. **b** HPLC chromatograms of electrolytes before and after 2 h GOR at 1 V under AM 1.5G one-sun illumination over NiOOH/ α -Fe₂O₃. It shows that formate is stable in the PEC reaction system.

Comment 4:

The Author claim that "the overall energy consumption of biomass-CO₂ photoelectrolysis is reduced by $32 \pm 2\%$, in comparison to the system operating with water oxidation" which does not take into account the energy need of biomass pretreatment, and the issues related to the generated chemical waste. What is the fate of the byproducts during the pre-conversion step? Apparently, the pretreatment is very important "control experiments show that the formate productivity without pretreatment is one or two orders of magnitude lower than those for pretreated samples" therefore it should be included in the complete picture.

Response:

We thank the reviewer for the comment. We agree with the reviewer that biomass pretreatment plays an important role in improving the formate productivity, which should be included in the overall process. Because it is difficult to quantitatively analyze the energy consumption of biomass pretreatment, we have revised the claim that "the overall energy consumption of biomass-CO₂ photoelectrolysis is reduced by $32 \pm 2\%$, in comparison to the system operating with water oxidation" to "the **cell voltage** of biomass-CO₂ photoelectrolysis is reduced by $32 \pm 2\%$ **with the introduction of pretreated biomass**, in comparison to the system operating with water oxidation" in the revised manuscript (*Page 13, Line 20*).

As for the fate of the byproducts (e.g., lignin) during the pre-conversion step, we consider that they can be separated and used for other applications. For example, lignin can be applied in composite materials and wood industries (Int. J. Biol. Macromol., 2020, 162, 985-1024).

Comment 5:

What is the reason for the performance decrease with time, and why does the performance restore after replacing the electrolyte? Is this a depletion in the sugar?

Response:

We appreciate the questions raised by the reviewer. The performance decreased with time was likely due to the coverage of catalytic active sites by the adsorbed organic intermediate formed

during the C-C bond cleavage process. The performance restoration was mainly attributed to the cleaning of photoelectrode surface that removed the adsorbed organic intermediates, while replacing the electrolyte played a minor role. Between every test, we thoroughly cleaned the photoelectrode surface with acetone, methanol and water under ultrasonication. There was only a slight depletion of sugar concentration in each cycle (~9% concentration decrease from 10 mM glucose). Such a depletion did not have a large influence on the photocurrent decrease. We have added the above discussions in the revised manuscript (*Page 11, Line 8*).

Comment 6:

There is a built in chemical bias because of the pH difference on the two sides. This also has to be taken into account during the energy efficiency calculations.

Response:

We thank the reviewer for the comment. A bipolar membrane that consists of an anion exchange layer laminated with a cation exchange layer was employed in our photoelectrochemical system. The utilization of bipolar membrane can maintain a steady-state pH difference between the two sides of anode and cathode, as has been demonstrated previously in the cases of water electrolysis and CO₂ electrolysis (ChemSusChem 2014, 7, 3021-3027; Nat. Energy 2017, 2, 17087; Acc. Mater. Res. 2021, 2, 1156-1166). Although pH difference can create a chemical bias on the two sides, there is also a built-in field formed within the bipolar membrane to exactly balance such a chemical bias difference between the two sides (J. Mater. Chem. A, 2015, 3, 19556-19562). Therefore, the potential or Gibbs free energy that required to perform the overall reaction will not change on the system level, which is regardless of the internal details of the system.

Comment 7:

The "unassisted" operation is also somewhat misleading. The term unassisted is generally used when a PEC cell alone can drive both half reactions. This is not the case here, as two additional Si solar cells are coupled to the system, if I understand correctly.

Response:

We thank the reviewer for pointing this out. We have removed the term "unassisted" in the revision.

Comment 8:

I know that it is not straightforward to do, but a solar to chemical conversion shall be estimated for the whole system. Ultimately this could show the merit of the presented approach.

Response:

We thank the reviewer for the comment. The solar to chemical conversion efficiency (η) has been calculated and added in the revision (*Page 14, Line 19, and Supplementary Note 1*). Due to the Faradaic efficiencies of formate production were different at anode and cathode, formate were produced based on two overall reactions as shown below. In both reactions, the anode reaction was glucose oxidation to formate. However, the cathode reactions were either CO₂ reduction to formate or H₂O reduction to H₂.

(Standard molar free energy of formation ΔG_f°): C₆H₁₂O₆ (s): -910.56 kJ/mol; CO₂ (g): -394.4 kJ/mol; H₂O (l): -237.13 kJ/mol; HCOOH (l): -361.3 kJ/mol; H₂ (g): 0 kJ/mol.

Overall reaction 1 (R1): $1/12 \text{ C}_6\text{H}_{12}\text{O}_6 + 1/2 \text{ CO}_2 + 1/2 \text{ H}_2\text{O} \rightarrow \text{HCOOH}$ $\Delta G_{R1}=30.345$ kJ/mol

Overall reaction 2 (R2): $1/6 \text{ C}_6\text{H}_{12}\text{O}_6 + \text{H}_2\text{O} \rightarrow \text{HCOOH} + \text{H}_2$ $\Delta G_{R2}=27.59$ kJ/mol

The solar to chemical conversion efficiency (η) was calculated according to the following equation:

$$\eta = \frac{N_{\text{formate}}^{R1} \times \Delta G_{R1} + N_{\text{formate}}^{R2} \times \Delta G_{R2}}{P_{\text{solar}} \times t}$$

where N_{formate}^{R1} and N_{formate}^{R2} are formate production amount via reaction 1 and 2, respectively, P_{solar} is the input solar power, and t is the reaction time. In the 80 h test, 1633.98 $\mu\text{mol}/\text{cm}^2$ and 227.27 $\mu\text{mol}/\text{cm}^2$ formate were produced via reaction 1 and 2, respectively. Thus, η was calculated to be $\sim 0.19\%$.

$$\eta = (1633.98 \times 30.345 + 227.27 \times 27.59) / (100 \times 80 \times 3600) = 0.194\%$$

Comment 9:

The comparison in Fig. 5f is totally unfair. The other studies involve different anode processes, than the present work. Thereby apples are being compared to oranges. Furthermore, the Figure legend "Performance comparison of this work with state-of-the-art unassisted PEC formate production from CO₂ and H₂O." is factually wrong, because "biomass" is also a reactant here.

Response:

We thank the reviewer for pointing this out. We are sorry for the misunderstanding due to the inappropriate presentation of the Figure and Figure legend. The Figure was intended to show the advantageous of current BOR-CO₂RR system over conventional OER-CO₂RR. Therefore, we have redrawn the figure to make it more clearly in the revision (*Page 30*), as shown in Figure R3 below. In addition, the Figure legend has been corrected as "Performance comparison of this work (BOR-CO₂RR) with state-of-the-art formate production from conventional OER-CO₂RR photoelectrolysis".

Figure R3. Performance comparison of this work (BOR-CO₂RR) with state-of-the-art formate production from conventional OER-CO₂RR photoelectrolysis.

Reviewer #2: (Comments to authors)

A novel scheme of biomass oxidation at anode coupled with CO₂ reduction at cathode (BOR-CO₂RR) using photoelectrochemistry to produce formate with Faradic efficiencies (FE) is reported. Non-precious 3d transition metal oxyhydroxide MOOH (M = Fe, Co, and Ni) supported on α -Fe₂O₃ photoanode and Bi/GaNi/Si-wafer photocathode was used. Authors have demonstrated a 32% reduction in energy consumption for BOR-CO₂RR compared to CO₂RR coupled with water oxidation. Plausible mechanism based on product analysis are postulated. DFT calculations are performed to support variation in photoanode activity among Fe, Co and Ni system.

This is an original study with proof-of-concept demonstration and conceptual design with potential for further development. The results are noteworthy, and authors have cited state-of-the-art developments in the field. Sufficient details of experimental studies are provided to reproduce the results. However, several issues remain to be addressed, notably morphological stability of photoanode (i.e., how to avoid leaching of NiOOH).

Response:

We thank the reviewer very much for the highly positive evaluation on the significance and originality of this work. We are very grateful to the reviewer for the encouraging and valuable comments and suggestions to improve the quality of the work. We have revised the manuscript accordingly, and our point-by-point responses can be found below.

We agree with the reviewer that the morphological stability of photoanode remains to be improved. To avoid the leaching of NiOOH cocatalyst, developing novel cocatalyst deposition methods or stable morphological structure with an additional coating (i.e., core-shell structure) are necessary in follow-up works. We have added the discussions in the revised manuscript (*Page 11, Line 17*).

Comment 1:

Brief DFT calculations presented in the manuscript doesn't necessarily establish high FE of Ni-anode system compared to Fe (FE of Ni-system is almost twice of Fe-system) as energy

values associated with intermediated and TSs for these two systems are too close.

Response:

We thank the reviewer for the comment. We agree with the reviewer that there is a gap between the DFT calculations and experimental observations, because the theoretical calculations are based on ideal atomic models and mainly simulate from the viewpoint of the thermodynamics, whereas the reaction kinetics may also play an important role in experimental measurements. However, the trends shown by the DFT calculations are consistent with that of the experimental observations. It is also worth noting that reaction rate and activation energy present exponential rather than linear relationship, as suggested by Arrhenius equation $k=A\exp(-E_a/RT)$. Therefore, a small difference in activation energy can result in a relevantly large difference in the reaction rate.

Comment 2:

Authors have demonstrated order of magnitude improvement in performance of BOR-CO2RR when biomass are pretreated. Pretreatment affects different biomass differently depending on composition. A suggested range of composition of substrate after pretreatment will be helpful to ensure target productivity from BOR-CO2RR.

Response:

We thank the reviewer for the comment. We agree with the reviewer that the composition of substrate after pretreatment affects the formate productivity. In this work, poplar sawdust, straw and bamboo were used as three representative biomass. The main compositions of these substrates before and after pretreatment were analyzed, as shown in Table R1 below. The compositions of substrate after pretreatment were monosaccharide sugars including glucose and xylose. It was found that there was a positive correlation between the formate productivity and sugar concentration. The pretreated sawdust solution showed the highest formate productivity due to the largest monosaccharide sugar concentration among the three samples. However, the proportion of glucose and xylose plays a minor role in the formate productivity, because the PEC performance of glucose and xylose are comparable, as shown in Figure R4 below. Therefore, the current method can be applicable to various lignocellulosic biomass substrates with different compositions of cellulose and hemicellulose.

Table R1. Compositional analysis of raw biomass substrates and the corresponding sugars solutions after pretreatment, and the formate productivity.

Raw biomass	Before pretreatment			After pretreatment ^[a]		HCOO ⁻ productivity (μmol cm ⁻² h ⁻¹)
	Cellulose (wt%)	Hemicellulose (wt%)	Lignin (wt%)	Glucose (g)	Xylose (g)	
Sawdust	58.12	19.00	16.53	0.193	0.051	18.4
Straw	40.02	24.93	4.81	0.145	0.067	17.3
Bamboo	49.13	21.58	20.05	0.163	0.066	17.4

^[a] The pretreatment was conducted by refluxing in H₂SO₄ solution at 100 °C for 5 h. Raw biomass feedstock (0.5 g), solution (100 ml).

Figure R4. LSV curves over NiOOH/ α -Fe₂O₃ using 10 mM glucose and xylose as the reactant substrates. All the experiments were conducted in Ar-purged 1 M aqueous solution of KOH (pH~13.6) under AM 1.5G one-sun illumination.

Comment 3:

Restructuring some of the languages as indicated in the annotated manuscript is suggested. Reviewer suggest accepting this manuscript for publication with minor edits/re-structuring as indicated above.

Response:

We are very grateful to the reviewer for the edits/re-structuring to improve the quality of the manuscript. We have carefully checked and polished the manuscript accordingly in the revision.

Reviewer #3: (Comments to authors)

The authors demonstrated a new photoelectrochemical system utilizing biomass oxidation reaction and CO₂ reduction reaction. Unifying the reaction product from both electrodes into formate was impressive and could appeal to the readers of the journal. However, some issues need to be addressed. Therefore, it is recommended to be accepted after major revision.

Response:

We thank the reviewer very much for the highly positive comments and recommendation of the work. All of the comments made by the reviewer are helpful for improving the quality of our work. Our detailed, point-by-point responses to the reviewer’s comments can be found below.

Comment 1:

The current title and abstract don't include information on the materials used or, in this case, systematic details. As Nature Communication aims to present important advances of significance to specialists within each field, the title and abstract should be more specific.

Response:

We thank the reviewer for the comment. The title and abstract have been modified as per the reviewer’s comment as follows. We have revised the title to “Renewable formate from sunlight,

biomass and CO₂ in a photoelectrochemical cell". In addition, we have added the following sentence of photoelectrode material information in the Abstract of the revision (*Page 2, Line 5*). Non-precious NiOOH/ α -Fe₂O₃ and Bi/GaNi/Si wafer were used as photoanode and photocathode, respectively.

Comment 2:

Though the Solar-to-fuel or solar-to-chemical efficiency is not calculated in this article, an estimation over figure 5 shows that comparatively low STF was achieved in this article. Recently an STF value exceeding 11% was reported. Refer to small, 2021, 17, 29, 2101128. Please add calculated STF values to the manuscript. Was the PV-PEC system optimized to achieve the best performance of formate production?

Response:

We thank the reviewer for the comment. The solar to chemical conversion efficiency (η) has been calculated and added in the revision (*Page 14, Line 19, and Supplementary Note 1*). Due to the Faradaic efficiencies of formate production were different at anode and cathode, formate were produced based on two overall reactions as shown below. In both reactions, the anode reaction was glucose oxidation to formate. However, the cathode reactions were either CO₂ reduction to formate or H₂O reduction to H₂.

(Standard molar free energy of formation ΔG_f): C₆H₁₂O₆ (s): -910.56 kJ/mol; CO₂ (g): -394.4 kJ/mol; H₂O (l): -237.13 kJ/mol; HCOOH (l): -361.3 kJ/mol; H₂ (g): 0 kJ/mol.

Overall reaction 1 (R1): $1/12 \text{ C}_6\text{H}_{12}\text{O}_6 + 1/2 \text{ CO}_2 + 1/2 \text{ H}_2\text{O} \rightarrow \text{HCOOH}$ $\Delta G_{R1}=30.345$ kJ/mol

Overall reaction 2 (R2): $1/6 \text{ C}_6\text{H}_{12}\text{O}_6 + \text{H}_2\text{O} \rightarrow \text{HCOOH} + \text{H}_2$ $\Delta G_{R2}=27.59$ kJ/mol

The solar to chemical conversion efficiency (η) was calculated according to the following equation:

$$\eta = \frac{N_{\text{formate}}^{R1} \times \Delta G_{R1} + N_{\text{formate}}^{R2} \times \Delta G_{R2}}{P_{\text{solar}} \times t}$$

where N_{formate}^{R1} and N_{formate}^{R2} are formate production amount via reaction 1 and 2, respectively, P_{solar} is the input solar power, and t is the reaction time. In the 80 h test, 1633.98 $\mu\text{mol}/\text{cm}^2$ and 227.27 $\mu\text{mol}/\text{cm}^2$ formate were produced via reaction 1 and 2, respectively. Thus, η was calculated to be ~0.19%.

$$\eta = (1633.98 \times 30.345 + 227.27 \times 27.59) / (100 \times 80 \times 3600) = 0.194\%$$

We agree with the reviewer that the solar to chemical conversion efficiency of current system is lower than state-of-the-art reports, although our system has been optimized to achieve the best performance of formate production. Currently, the formate productivity of our system is largely limited by the low performance of the photoelectrodes. Further improvement of the performance is anticipated by integrating better photoelectrodes with higher photocurrent, such

as BiVO₄ or SnS mentioned by the reviewer. The related discussions have been added in the revised manuscript (*Page 14, Line 20*).

Comment 3:

The faradaic efficiency (maximum 85.2%) of the photocathode is relatively low when compared with other reported performances of Bi/GaN NW electrodes even after cycle optimization, as shown in Figure S31. Maybe the 20-cycled sample should be tested at higher potentials.

Response:

We thank the reviewer for the comment. We agree with the reviewer that the faradaic efficiency (maximum 85.2%) of the photocathode is not the highest among the report values. The tests of 20-cycled sample at higher potentials have been performed, which however did not show improved performance. Further investigations of other Bi deposition methods (e.g., thermal evaporation) are underway in our lab to enhance the performance.

Comment 4:

The X-axis in figure S31(b) is wrongly labeled.

Response:

We thank the reviewer for pointing this out. We have corrected the X-axis label to “Cycle numbers” as shown in Figure R5 below in the revision.

Figure R5. FEs of different products and j_{HCOO^-} of Bi/GaN/Si photocathodes with varied electrodeposition cycles of Bi.

Comment 5:

Is there any reason why hematite and GaN/Si were used as photoelectrode materials? There are several more candidates with higher photocurrent, such as BVO or SnS. Refer to the following [1] Energy Environ. Sci., 2022,15, 672-679 [2] Adv. Sci., 2021, 8, 21, 2102458 [3] ACS Appl. Mater. Interfaces 2020, 12, 13 [4] Solar RRL, 2019, 3, 12, 1900301

Response:

We thank the reviewer for the comment. Hematite and Si were chose as the photoelectrode

materials in this work due to their low cost, earth abundance, well-established performance, and suitable bandgaps of 2.1 and 1.1 eV as top and bottom light absorber, respectively. Although the two materials were used for the proof-of-the-concept demonstration, it is expected that the performance of our tandem PEC system can be further improved by using other candidate materials with higher photocurrent at favorable potentials, such as BiVO₄ and SnS. We have added the related discussions and cited the relevant references mentioned by the reviewer in the revised manuscript (*Page 14, Line 20*).

REVIEWERS' COMMENTS

Reviewer #2 (Remarks to the Author):

Thank you for considering the review comments and updating the manuscript accordingly. I believe the updated manuscripts will provide a better understanding of this study to future readers. I recommend accepting this manuscript for publication to Nature Communications.

Reviewer #3 (Remarks to the Author):

Many parts were revised accordingly by the authors. However, an issue regarding the calculation of solar-to-chemical efficiency remains to be addressed. The values of $N(R1)$ and $N(R2)$ were 1634 and 227, respectively, without units. How did the authors determine those values? Faradaic efficiencies for formate and H_2 in the cathodic compartment along the 80 hours must have been taken. It is recommended to add the data in the supplement.

Point-by-point response to the reviewers' comments

Title: “*Renewable formate from sunlight, biomass and CO₂ in a photoelectrochemical cell*”

(Manuscript ID: NCOMMS-22-36755A)

We sincerely thank again all Reviewers for their valuable comments and suggestions, which are certainly helpful in improving the quality of our work. We have carefully and systematically responded to all the points raised. The Reviewers' comments are in blue fonts and our responses in black fonts. We have also highlighted the revised text in yellow in the main text. Provided below are our detailed responses to each point.

Reviewer #2: (Comments to authors)

Thank you for considering the review comments and updating the manuscript accordingly. I believe the updated manuscripts will provide a better understanding of this study to future readers. I recommend accepting this manuscript for publication to Nature Communications.

Response:

We thank the reviewer very much for the highly positive comments and for the acceptance of our work. We highly appreciate the reviewer again for the valuable time and encouraging comments in improving the quality of this manuscript.

Reviewer #3: (Comments to authors)

Many parts were revised accordingly by the authors. However, an issue regarding the calculation of solar-to-chemical efficiency remains to be addressed. The values of N(R1) and N(R2) were 1634 and 227, respectively, without units. How did the authors determine those values? Faradaic efficiencies for formate and H₂ in the cathodic compartment along the 80 hours must have been taken. It is recommended to add the data in the supplement.

Response:

We are very grateful to the reviewer for the valuable comments and suggestions, which enormously improve the quality and clarity of this manuscript. Provided below is our detailed response to this point.

The total amount of formate produced at the anode and cathode sides after 80 h test were detected to be 1044 $\mu\text{mol cm}^{-2}$ and 817 $\mu\text{mol cm}^{-2}$, respectively, as shown in Figure R1 below. Twice the amount of formate produced at the cathode can be regarded as formate produced via R1 reaction (1634 $\mu\text{mol cm}^{-2}$). Besides, formate produced at anode side is higher than that of cathode side and the extra formate at anode was regarded as produced via R2 reaction (227 $\mu\text{mol cm}^{-2}$). We have added the above discussions in the revision (Supplementary Figure 35, Supplementary Note 1).

Figure R1. Amount of formate produced at photoanode and photocathode in 80 h test.